# An Efficient Healthcare Data Mining Approach Using Apriori Algorithm: A Case Study of Eye Disorders in Young Adults

**Kanza Gulzar** [1,*] **, Muhammad Ayoob Memon** [2] **, Syed Muhammad Mohsin** [3,4] **, Sheraz Aslam** [5,6,*] **, Syed Muhammad Abrar Akber** [7] **and Muhammad Asghar Nadeem** [8]

1 University Institute of Information Technology, PMAS Arid Agriculture University, Rawalpindi 46000, Pakistan
2 Internal Medicine Department, Jinnah Sindh Medical University, Karachi 75510, Pakistan
3 Department of Computer Science, COMSATS University Islamabad (CUI), Islamabad 45550, Pakistan
4 College of Intellectual Novitiates (COIN), Virtual University of Pakistan, Lahore 55150, Pakistan
5 Department of Electrical Engineering, Computer Engineering and Informatics, Cyprus University of Technology, Limassol 3036, Cyprus
6 Department of Computer Science, CTL Eurocollege, Limassol 3077, Cyprus
7 Department of Computer Graphics, Vision and Digital Systems, Faculty of Automatic Control, Electronics and Computer Science, Silesian University of Technology, 44-100 Gliwice, Poland
8 Department of Computer Science and IT, University of Sargodha, Sargodha 40100, Pakistan
* Correspondence: kanza@uaar.edu.pk (K.G.); sheraz.aslam@cut.ac.cy (S.A.)

**Abstract:** In the public health sector and the field of medicine, the popularity of data mining and its usage in knowledge discovery and databases (KDD) are rising. The growing popularity of data mining has discovered innovative healthcare links to support decision making. For this reason, there is a great possibility to better diagnose patient's diseases and maintain the quality of healthcare services in hospitals. So, there is an urgent need to make disease diagnosis possible by discovering the hidden patterns from the patients' history information in developing countries. This work is a step towards how to use the extracted knowledge to enhance the quality of healthcare facilities. In this paper, we have proposed a web-centered hospital information management system (HIMS) that identifies frequent patterns from the data with eye disorder patients using the association rule-based Apriori data mining technique. The proposed framework has the capability to overcome all the key issues and problems in the current hospital information management system regarding data analysis and reporting services. For this purpose, data were collected from more than 1000 university students (China citizens) both online and manually (printed questionnaire). After applying the Apriori algorithm on the collected data, we revealed that almost 140 individuals out of 1035 had myopia (near-sighted disorder), at current age of 22 years, and that there were no male patients found with myopia. We concluded that their clinical relevance and utility can generate favorable results from prospective clinical studies by mapping out the habits or lifestyles that potentially lead to fatal diseases. In the future, we plan to extend this work to fully automate HIMS to help practitioners to diagnose the reasons of various diseases by extracting patient lifestyle patterns.

**Keywords:** data mining; knowledge discovery and databases (KDD); hospital information management system (HIMS); diagnosis; pattern discovery; health care; medical data; patient record





## 1. Introduction

Today, data mining has prevailed due to its promising applications. It has become a valuable technique for disease diagnosis in the healthcare sector. More effective expert decision-making is made possible by analyzing accessible data on a particular problem. The key objective of data mining is to extract vital information from a large volume of raw data and is used to classify and analyze medical data [1]. Healthcare scholars and researchers need a lot of data to validate their healthcare-associated activities with medicine

prescriptions that can capably recover patients from illness. In healthcare services, the role of information technology is remarkable [2].

The use of electronic medical records (EMR) by healthcare systems encourage the utilization of huge datasets holding relevant facts regarding a patient's life pattern. The impact of increased availability of panel electronic data on prospective detection of patterns for emerging diseases will definitely help in improving personalized care to ensure quality life. While we devise an HIMS, the history of patients including lifestyle, lab tests, interventions, diagnosis and recommendations are maintained in EMR. To find out the relevancy between a disease and the history for diagnosis, discovering frequent hidden patterns are potential key areas of importance in healthcare. Pattern discovery is the process of uncovering or mining the patterns from massive data sets [3]. National Research Council (NRC) defines pattern mining as a tool to identify/recognize terrorist activity; these patterns might be considered as minor/inaudible signals within a large dataset [4,5].

### 1.1. Motivation

Although there are many applications of data mining in daily life, some challenging issues still need to be seriously considered. The most frequently faced issue is pattern matching that implements the association rules to discover relations among variables in large sets of items, big datasets and huge business databases. Focusing on medical records, discovering causal relationships between events in patients, during their lifetime is the biggest issue in health care as the extraction of sequence of activities associated with a particular disease is difficult. Keeping this issue in mind, we have proposed this research work. As healthcare systems currently use electronic medical records (EMRs), this promotes the use of health information from large data sets. In practice, initially, hospital information systems (HISs) in developing countries are not as common, if at all, providing limited facilities. These key issues are mainly related to data privacy, security, complexity, data integrity, and absence of unique national patient identifier in many third world countries. Another challenging issue in health care systems is accessibility of data as a large amount of data facts are emerging out from different sources such as laboratories, clinics, hospitals, and primary care agencies. There are different ways to store patient's health record, such as individual databases and the traditional paper form, which is a major reason for the loss of evidence for disease diagnosis. As data are available in various formats and is scattered, it becomes difficult to handle and to extract useful information from it. Another major problem with today's HIS used in developing countries is that they do not facilitate the clinical data analysis. Clinical data analysis plays major role in diagnosis, treating or preventing disease, and can track efforts taken in quality of care improvements.

### 1.2. Contribution

In this study, we propose an association rule-based hospital information management system (HIMS) for discovering frequent patterns of data in clinical databases of eye hospitals. For this purpose, an Apriori algorithm of association rule is adopted. The research gap and rationale of the study is discussed with references from the literature in Section 2.1. This study is a module of our ongoing project for our central hospital. The objective of the current study was the extraction of frequent patterns of patient lifestyle (hobbies) to detect and find out the relevancy among the disease and the hidden lifestyle patterns relationship.

The proposed web-based HIS framework has the capability to resolve the above-stated problematic key issues. The proposed work is validated through a case study of eye disorders (EDCS) in which the frequent patterns are identified from a large clinical dataset using association rule-based Apriori algorithm. The benefits of this potential research are twofold: on the one hand, it will handle data control issues, and on the other hand, it will help the practitioners to find out the underlying facts that can cause eye disorders in teenagers. The association rule mining is used to extract information from clinical data for visualizing the correlation of diseases and the habits of the students. This way, doctors can benefit from the products of this proposed system by knowing which combination of

certain habits can lead to weakening of eye sight. At first, we took hobbies that are mostly considered the probable reason of weak eyesight as a trial data. This will help us in the future to detect secondary effects, new counter clues, and unseen interactions between the events of medical records by extracting frequent patient behavior.

The rest of the paper is organized as follows: Section 2 presents the work related to the method used to achieve study's objective. Section 3 explains the proposed model of the HIMS and frequent patterns are discovered using the Apriori algorithm in Section 4. Section 5 relates to experimental setup and results, Section 6 is related to discussion. Finally, the Section 7 concludes this study.

## 2. Literature Review

A lot of research is carried out in the data mining field for frequent pattern discovery in the medical field. Rajkumar et al. [6] have defined the health care information system, which needs a logical method of extracting knowledge. They have also applied associative rules on the healthcare dataset. The results can be evaluated in terms of accuracy, precision, and recall.

Shreenath and Madhu [7] proposed a prediction system for the placement of students into respective groups to help the institutes to extract some relevant facts that could be investigated for the selection of a future plan of study. The system performed prediction on the basis of the historical information stored in a database through the association rule using the Apriori algorithm. The goal is to handle key issues of equity and access. The need is to restructure the system to address the emerging challenges to build an effective, accountable, and self-improving system [8]. In [9], the authors have conducted surveys to explore the value of several data mining techniques, such as classification, association, clustering, and regression in the health domain. They also have identified future challenges to data mining in healthcare. In the research work [10], the authors have compared the impact of various data mining techniques, approaches, and tools for the healthcare sector. They have also applied various types of data mining applications on the health care record.

In the paper [11], the authors adopted a purely data-driven approach to patient segmentation with application for hip fracture care in Ireland. They grouped the elderly patients on the basis of certain features such as length of stay, similarity of age, and elapsed time to surgery through K-means clustering. They used clustering analysis to determine coherent clusters of patients, the correlations that may probably exist in relation to characteristics of patients, their outcomes, and care-related factors. Simply, we can say that this study investigates the possible influence of one feature or a factor on another within the explored clusters of patients.

Pellicer-Valero et al. [12] used the idea of profiling groups of patients in clusters. According to them, a deep understanding of population characteristics through profiling groups of patients in clusters aids in the identification of people with cornification and evolving the appropriate corrective measures. Their work is impressive in a sense to adopt the clustering of the lifestyle activities that identify different groups of patients with disease for common therapeutic measures. A process mining classical-paradigm-based pattern recognition algorithm is presented in [13]. This algorithm has the capability to infer clinical pathways to aid the clinical pathways designs. Kai-Ming Jhang et al. [14] investigated the care needs for people suffering from dementia and worked to reveal precise combinations of care needs for such patients. They claimed that the idea of the bundling of necessary care needs proved to be more effective for patients with dementia on an extended level.

There are many data mining applications in the business sector. Wu [15] showed that for a huge business, we can understand the customer's consumption habits through mining association rules. The knowledge gained can be used for effective market prediction, market analysis, and personalizing the services for the selected customers to increase the sales efficiency and market competitiveness. He has used the Apriori algorithm for this purpose. Haoyu Xie [16] briefly described the basic concepts of data mining, association rules, and the pros and cons of the Apriori algorithm. The drawback of the Apriori algorithm is that,

for a larger dataset, it takes more memory, which creates a burden and consumes more time, which reduces the efficiency of the algorithm. However, Haoyu Xie emphasized the need to understand the basics of data mining and association rules to better understand Apriori algorithms for prediction and decision support. He agrees with Wang's [17] view that the Apriori algorithm reduces the coding difficulty for the programmer because it offers less space complexity. The nature of the algorithm has advantages, such as pruning processes that allow avoiding many repetitive operations, which helps to improve the speed of the algorithm. Moreover, the joining and pruning is simple and easy to implement on large item sets. We are convinced to use the Apriori algorithm for our proposed work because of Haoyu's assenting review.

*2.1. Research Gap*

In the above section, we briefly reviewed how data mining helps in various fields of daily life. There are two reasons we are conducting this study: (i) We conclude from the literature that applying the concept of data mining in e-commerce places the emphasis on customer's consumption habits through mining association rules. Similarly, this can be implemented in the medical field for the diagnosis process by extracting patient histories; however, in medical diagnosis, more focus is placed on mining treatment patterns from event logs. On the other hand, the diversity of the behavior data is not well structured. This lack of diagnosis of the underlying causes of diseases in many patients, together with their lifestyles, consumption of different kinds of food stuffs, and residence commodities, make it uncertain whether a disease is linked to a particular habit or not. We can say that patients may be likely to have many diseases because of their habits, lifestyles, and parental histories. However, it is unknown which disease is prompted by which factor because their identified value is missing in the HIMs. (ii) There are many applications established in developed countries and are producing wonderful results; however, still, the concept of patient lifestyle history relating to disease is left to be incorporated in HIS. At the same time, for a country such as Pakistan, these are newer, and it is harder to implement such systems in general because, on one hand, the lifestyle history is not considered mandatory for disease diagnosis, and on the other hand, the patient is also reluctant to share all of their information due to privacy issues. Taking a note of the above-mentioned gap, we have tried to propose an HIS model that permits the identification of the habits and lifestyle patterns in the health system that are associated with a disease. This leads to the complete treatment of the disease by discovering the common lifestyles followed by patients with a particular diagnosis. This is a pilot study in which we took examples of eyesight problems in a real scenario. The reason for considering this eye disorder is that we can easily find students with this problem to maintain a database. We have designed our research work keeping in mind the benefits of association rule mining and electronic HIS. The architecture of the proposed system is discussed below.

**3. Proposed Model of Hospital Information Management System (HIMS)**

The architecture used and the workflow of the proposed system are shown in Figure 1. The architecture used has three tiers. It is a deeply rooted software architecture, managing applications in three tiers facilitating users, managers, and programmers. Its key advantage is its independence of modules as it allows the user to enter and access the data through an application server and has no need to know the underlining process from the request entered to the retrieval of the data. The three-tier architecture has been the most prevalent architecture for decades and even today, modern tools using advanced technologies implement a three-tier architecture. It also provides the facility to store images in the database and have the ability to retrieve the imagery data or information updated by the doctors and the patients. ImageJ, which is an image processing application, is used for fetching images from the picture storage server. The 8-bit, 16-bit, and 32-bit images can be displayed, edited, analyzed, processed, and printed by ImageJ [18].

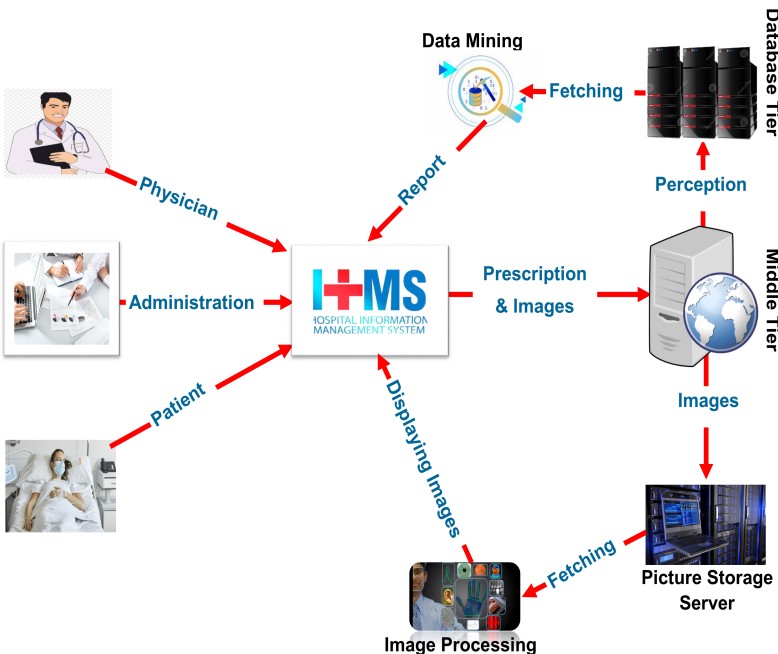

**Figure 1.** Workflow of proposed hospital information management system (HIMS).

A data mining tool was implemented in order to find frequent patterns of a disease from the patient's health record. Mining association rules are used to distinguish relative associations between datasets in databases. The discovery of patients' lifestyle patterns help the practitioner to make the right decision for diagnosis that will ultimately lead to the correct treatment. If we consider the current HIMS, we will fail to find the impact of such patterns on the disease of patients, especially in Pakistan. Though, today, lifestyle is becoming a hot topic of consideration in the field of disease diagnosis, but its implementation in an appropriate manner has not yet been attained. In conclusion, developing a system with a logical approach will help to make decisions faster. For the medical field, it is employed to discover the most frequently occurring diseases.

The proposed framework uses datasets in comma-separated values file (CSV) format, whereas by default, the format is the attribute-relation file format (ARFF). Thus, the data set is converted from CSV format to ARFF format [19]. Administrators and doctors have the rights to access the data mining application portal. The patients can retrieve their own health information and have restricted rights. The function of the administrator module is to manage the patient actions and profiles, including their lifestyle activities, diet, and other habits (sleeping, playing, exercise, company, etc.) and medication records. In our case, we took only the habits that can cause eye disorders. By using this mode, the admin can easily add, delete, and update all of the information. Furthermore, the admin can view the overall reporting of a patient's data. With the help of the patient module, patients can also view, update, or add any new information in the existing profile; however, the admin can have all the data that were added before and deleted after editing by the patients. This helps the practitioners to dig out how the changing lifestyle affects the disease status. Patients are facilitated to share the medical records and their laboratory reports with an administrator and doctor by giving access to them. When a patient shares any information with a doctor, it will be displayed in that particular doctor's module as well as to any other doctors that will examine that particular patient by using the unique patient id. Once any patient case is displayed in the module, doctors can examine the patient's information and can suggest if any medicines or laboratory tests, etc., are required. Later, the details can be stored in the SQL server database.

## 4. Mining Patient's Health Records to Identify Frequent Patterns Using Association Rule-Based Apriori Algorithm

Mining patient's health records for the identification of frequent patterns is an important medical investigation area. The research community has evaluated various data mining techniques. The authors of [20,21] have discussed several data mining algorithms used for data preprocessing, classification, and clustering. They have provided a detailed comparison of the discussed techniques and also elaborated with respect to industrial applications. After a comprehensive review of the clinical data preprocessing and analysis techniques, we were able to develop our proposed data analysis technique, which is discussed in the following section.

### 4.1. Data Analysis

The data extracted from the EMR were organized through data preprocessing. Initially, the transformation of raw data was performed through recorded patient identifiers to obtain the information of the hobbies to be analyzed. Different phases of the knowledge discovery process proved to be a well-structured and effective pattern extraction mechanism from the database. It comprises inspecting, cleaning, transforming, and modeling the data to discover useful information, suggesting remarks or conclusions, and supports decision making. The knowledge discovery process is described below and shown in Figure 2.

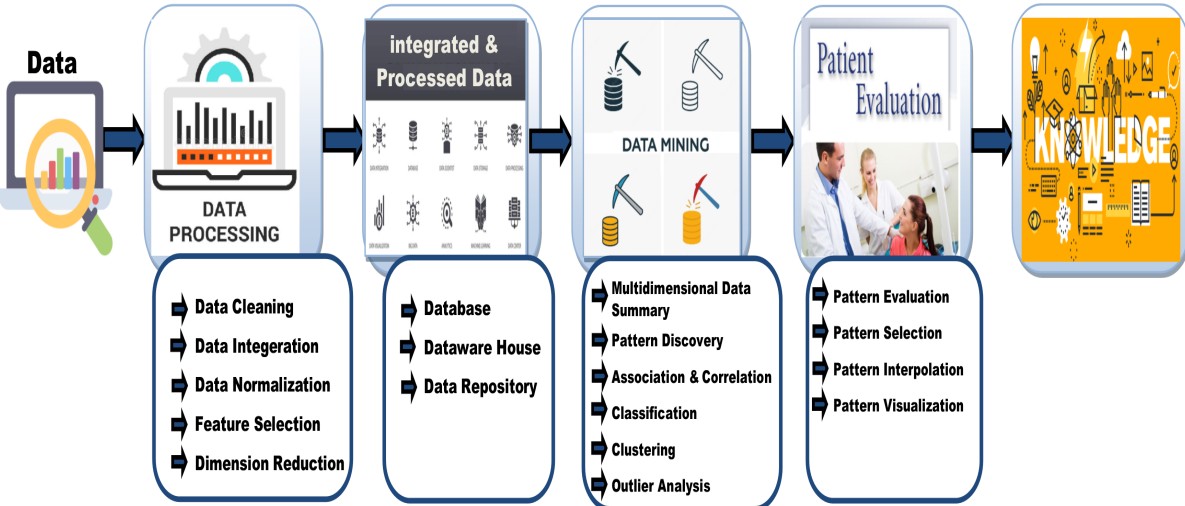

**Figure 2.** The knowledge discovery process.

I.  *Data Requirement:* Data requirement is the first and most important phase that defines what kind of data are required for analysis purposes. It varies from company to company and from requirement to requirement for analyzing a business process. Our data collection relates to eye disorder patients. The data obtained can be numerical or categorical (numerical or text). For our case study, we asked Chinese students about their age, sex, location, profession, hobbies, disease (if they have any) and age at the time of diagnosis, and use of glasses or treatments to overcome a disease.

II. *Data Collection:* For data collection purposes, a questionnaire containing 10 questions relevant to the data requirement was created. The questionnaire contains "input data, single selection and multiple selection" type questions. An online survey form is created on Facebook, What's App, and We Chat to achieve this goal.

III. *Data Processing:* It is the process of organizing or handling the collected data for analysis purposes. There is a verity of data mining tools used for data processing purposes. In this research work, Rapid Miner5 community edition is used for most of the data mining tasks.

IV. *Data Cleaning:* Data cleaning is the process of preventing and correcting data-related errors. We found that, in the eye disorder patient's records, some of the individuals

did not fill out required information, which resulted in missing values. These errors were removed using data mining tools such as Rapid Miner5.

V.  *Analyzing Data:* A variety of data mining tools (either commercial or open source) are present to analyze data. Many tools provide different kinds of charts for data analysis purposes. The data of eye patients are analyzed by using Bar, Scatter, and histogram charts.

VI.  *Discovering Frequent Patterns:* After analyzing the data of eye patients, we found interesting patterns by using the association rule approach. The Apriori algorithm, using an association rule, was employed to accomplish this goal.

In the literature, an extensive study was conducted to evaluate data mining techniques. There are several data mining algorithms that have been employed for data preprocessing, classification, and clustering. A comparison of the pros and cons of these techniques in the practical applications is also elaborated for their use in industry. We have conducted an extensive review of the methods used for preprocessing and analyzing clinical data studied [20,21], and concluded that a combination of different data mining techniques can work as a productive tool.

*4.2. Apriori Algorithm*

Frequent pattern mining using the Apriori algorithm was also used to mine the associations between coexisting diseases. Agrawal [22] used the Apriori algorithm to mine rules in a customer transaction database. Apriori is more popular algorithm used to find all frequent item sets and association rule learning. The main reason to use the Apriori algorithm is that it is well understood, easy to implement, has many derivatives, and is one of the top ten classic algorithms that has proven to be the best in data mining [23]. It uses a "bottom up" approach, in which all the frequent subsets stretched an individual item at a time (the step is known as candidate generation), and then groups of generating candidates are tested against the data. The algorithm comes to an end when no further successful extensions are found or, in other words, no further candidates can be generated from the data. The Apriori algorithm uses a breadth-first search (BFS) and a hash tree structure to count the candidate item sets effectively and efficiently. The Apriori algorithm generates candidate item sets of length "n" from item sets of length "n − 1". Finally, the Apriori algorithm prunes the generated candidates that have an infrequent sub pattern. The comprehensive pseudocode of the Apriori algorithm is shown in Algorithm 1.

---

**Algorithm 1** Pseudo code of the Apriori algorithm.

---

m: Patients' Hobbies data item set of size m
Fm: Frequent item set of size m
m: = 1
Fm: = frequent items;
**While** (Fm!= $\phi$)
**do** {
Cm + 1: = Patients' Hobbies data generated from
Fm;
Derive Fm by counting Patients' Hobbies data in
Cm + 1 with respect to DB at minsup;
m: = m + 1
}
**Return** Fm

---

*4.3. Apriori Algorithm on Patients Hobbies Record*

One of the most important questions asked in the eye disease patient's questionnaire was related to the hobbies or interests of individuals. This question is asked to figure out the relationship between the individual's hobbies and their eye disorder (here, we examined refractive errors, also known as near or far sightedness). They were given five predefined

options (can multi select) for hobbies/interests that can cause this kind of disorder (near or far sighted). The options are books reading in soft form, books reading in hard form, watching movies, playing games on mobile and work station. The fact to be remembered is that the other aspects, such as genetic factors, are not discussed in this work. We have developed this model to explore the results at a restricted level. The effort is made to find the frequent patterns for the individual's hobbies to dig out the most popular hobby of the individuals with eye disorders. The hobbies or interests taken into consideration are listed in Table 1.

**Table 1.** Hobbies found in the data with eye disorder patients.

| Sr. # | Abbreviation | Description |
| --- | --- | --- |
| 1 | BRHF | Books Reading in Hard Form |
| 2 | WM | Watching Movies |
| 3 | PGW | Playing Games on Work Station |
| 4 | BRSF | Books Reading in Soft Form |
| 5 | PGM | Playing Games on Mobile |

The data collected from more than 1035 individuals against the question "what are your hobbies/interests?" contains 29 distinct sets of answers. Some of the people have just one hobby; some have two or more hobbies as can be seen in Table 2. From these twenty-nine distinct answer sets, an effort is carried out to find the Freq. 1-item-sets and Freq. 2-item-sets for the hobbies of individuals using the Apriori algorithm. All the frequent item sets are determined by using association rules based on the Apriori Algorithm and all the answers and pair of answers are represented in Table 2. Each transaction in the database is taken and observed as a set of item (an item set). Given a threshold, the Apriori algorithm recognizes item sets that are subsets of at least transactions in the database [19].

The entire dataset was scanned to acquire A = Books Reading in Hard Form, Watching Movies, Playing Games on Work Station, Books Reading in Soft Form, Playing Games on Mobile. At this stage, all the gained data were implemented as candidate set 1, named C1. The first scan was carried out on the available data of hobbies. After obtaining C1, the support of each item was mapped with the minimum support (min sup). The items having support greater than the min sup were placed in this category; other items having less support were pruned. Finally, frequent item-sets F1 were found. Figure 3 shows the candidate set C1 and frequent item-set F1.

Then, the items left by item-set F1 were used to perform the connection operation f1 f2, and candidate set C2 was generated from the frequent item-set F1. The candidates were taken in a pair of two, and their support was calculated. The second scan then took place, and scanned the whole data set again, counting the data of two projects at the same time, then calculating the degree of support to form candidate set 2 (C2). A comparison of min_support is made to quantify the item set after pruning to obtain frequent item-set F2. The association connection of candidate set C2 and frequent item-set F2 are also represented in Figure 3.

**Table 2.** Different hobby item sets found in data with eye disorder patients.

| Sr. # | Hobby | Count |
| --- | --- | --- |
| 1 | PGW | 70 |
| 2 | PGM | 10 |
| 3 | PGM, PGW | 5 |
| 4 | BRHF, PGW | 35 |
| 5 | BRHF | 240 |
| 6 | BRHF, PGM | 5 |

**Table 2.** *Cont.*

| Sr. # | Hobby | Count |
|-------|-------|-------|
| 7 | BRHF, PGM, PGW | 5 |
| 8 | BRHF, WM | 105 |
| 9 | BRHF, WM, PGW | 10 |
| 10 | BRHF, WM, PGM | 15 |
| 11 | BRHF, WM, PGM, PGW | 5 |
| 12 | BRSF | 60 |
| 13 | BRSF, PGW | 20 |
| 14 | BRSF, PGM | 5 |
| 15 | BRSF, PGM, PGW | 10 |
| 16 | BRSF, BRHF | 40 |
| 17 | BRSF, BRHF, PGW | 15 |
| 18 | BRSF, BRHF, PGM | 15 |
| 19 | BRSF, BRHF, PGM, PGW | 35 |
| 20 | BRSF, BRHF, WM | 55 |
| 21 | BRSF, BRHF, WM, PGM | 5 |
| 22 | BRSF, BRHF, WM, PGM, PGW | 25 |
| 23 | BRSF, WM | 25 |
| 24 | BRSF, WM, PGW | 10 |
| 25 | BRSF, WM, PGM | 20 |
| 26 | WM | 130 |
| 27 | WM, PGW | 30 |
| 28 | WM, PGM | 20 |
| 29 | WM, PGM, PGW | 10 |

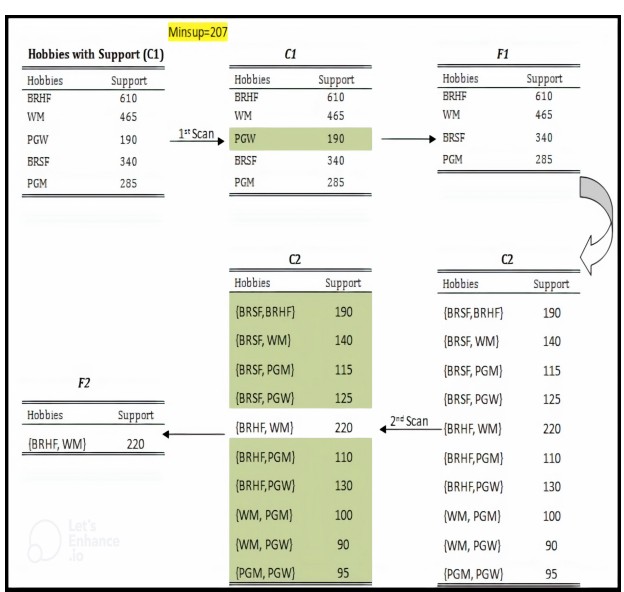

**Figure 3.** Candidate set C1 and frequent item-set F1 generating Candidate set C2 and frequent item-set F2.

## 5. Experimental Setup and Results

After collecting, cleaning, and processing, data with eye disorder patients were analyzed by using bar, graphs, and scatter charts in Rapid miner, a data mining tool. Now, we examine the organized, processed, and error-free data of eye patients in terms of charts. The chart shown in Figure 4 clearly shows that the number females is much more than the number of males. A large number of male and female individuals responded to the survey and a few reported to have myopia (nearsightedness); the rest of the individuals were not suffering from any eye disorder.

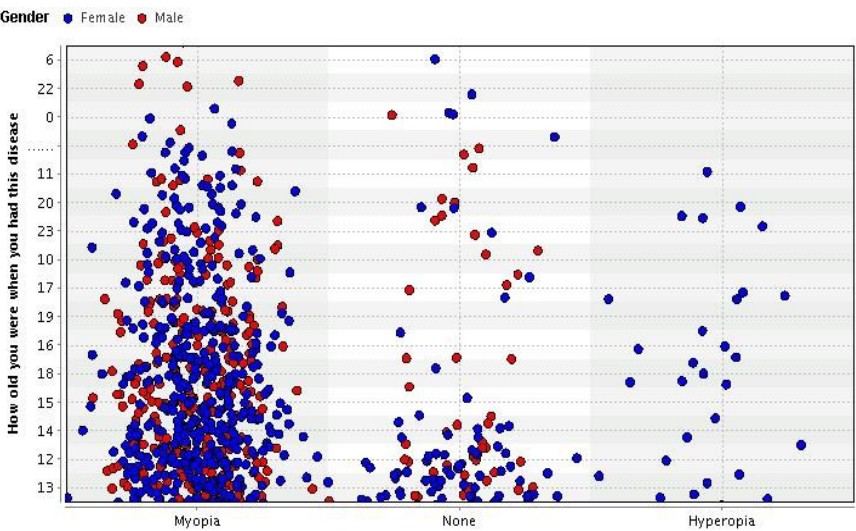

**Figure 4.** Scatter chart of eye diseases vs. age.

An equal number of males and females were taken into account for sampling, but the remarkable point that catches our attention is that there are a few myopia (nearsightedness) cases and it is shown that only female individuals had hyperopia, and no male was found with myopia. The reason may be that boys have many activities other than the described hobbies, such as playing football, running, and gym, which are part of healthier lifestyles. Figure 5 shows a histogram of the eye disorder count with respect to the current age of the individual. The histogram reveals information that shows that almost 140 individuals out of more than 1035 have myopia at the current age of 22.

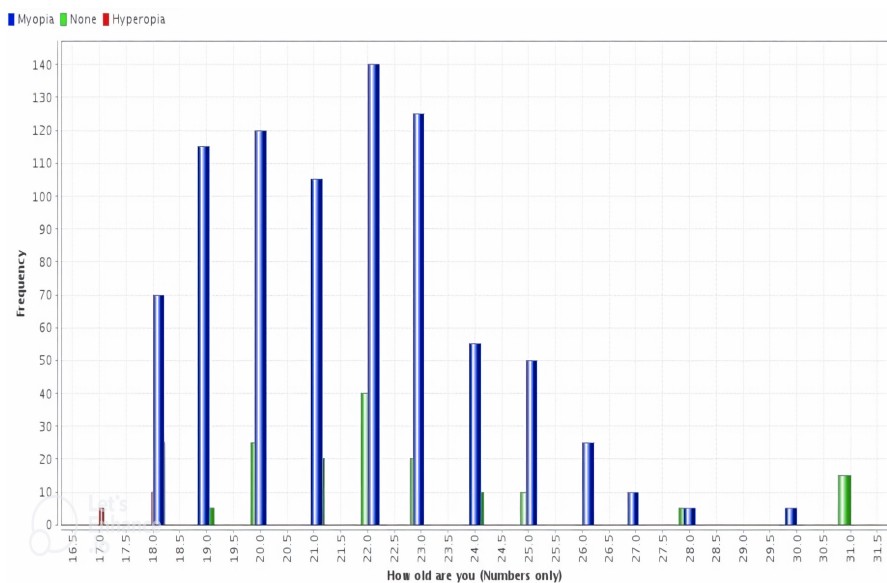

**Figure 5.** Histogram for current age and diseases.

After visualizing the data of eye disorder patients, we are interested in finding the frequent pattern in data. For this purpose, TANAGRA was employed [24]. TANAGRA is another open source data mining tool used for data analysis and pattern discovery purposes. The Hobby data set was passed through the Apriori algorithm of TANAGRA. Figure 6 indicates that there is only one frequent item-set found using the Apriori algorithm. It verifies the result discussed in Section 4.

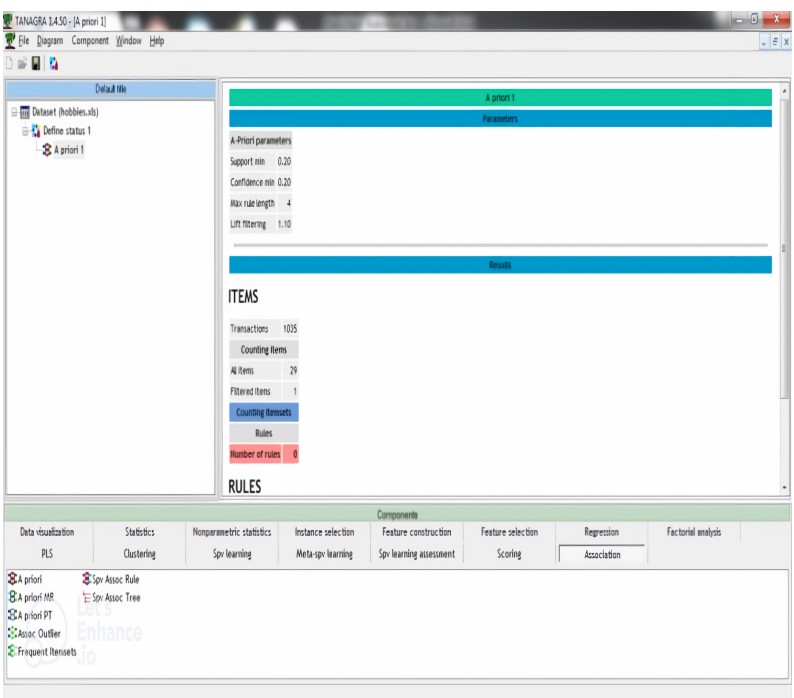

**Figure 6.** Apriori algorithm result in TANAGRA.

## 6. Discussion

Data mining techniques require an approach that provides a promising way to visualize data from a new angle that permits us to discover hidden, valuable, and applicable patterns. The purpose of our proposed system is to extract useful patterns from a database that has been created by taking information from different university students. To develop our idea of exploring hobbies for a particular eye disorder in HIMS, we gathered data such as the data, doctors and assistants collect from their patients regarding the disease for which they are complaining and then take the patient's history. Thus, the database of our proposed system was formed. The collected data are unstructured, so pre-processing of that data was performed before applying the Apriori algorithm to mine for frequent patterns. In different hospitals, patients are treated for a particular disease according to their medical record. In this paper, as we have used a case study with eye disorders, after identifying the observed patterns (hobbies) of eye disorder patients, this approach can be used for future datasets to provide appropriate guidance and counseling to young adults.

An example of recommendations made to young adults from patterns extracted:

1. Encourage young females (as they are more prone to eye problems) to perform some physical exercise.
2. Convince them to visit some fun places instead of staying in the hostel for long to avoid more frequent use of mobiles.
3. Ensure the physical participation in campus activities such as games.
4. Introduce them to healthy meal plans full of vitamins essential for eye health.

Likewise, many other outcomes can be formulated and measures taken for different combinations of patterns associated with a particular disorder.

The experimental evaluation of HIMS was integrated into the TANAGRA tool, featuring five hobbies (BRHF, WM, PGW, BRSF, and PGM), age of the students and the presence of the disorder. From the outcome of the research, we can expect to execute the proposed framework on large databases with improved performance. Here, we have introduced the concept of the lifestyle patterns that may cause to have some kind of eye disorders. At first, we took hobbies of young adults as a trial data. We have tested the proposed approach for mining hobbies linked with eye disorders on data gathered from different university students. The results of our approach show that the technique we used can provide a variety of feedback to improve existing HIMS. Similarly, the same can be implemented to any other database to obtain promising results for any other organization. Our implementation demonstrated the applicability of the recognized model to improve healthcare services. With a few modifications in the future, the system will be able to work similarly for certain infectious diseases and interact with input datasets to generate correlations for better prevention, diagnosis, and treatment.

The results of our work ensure accurate and fast processing of medical data and symptoms. There are several studies that support our work. In a survey [25], it was found that the mining of required information from large amounts of raw data plays an important role in medical science, including disease diagnosis, decision making, and drug development, each of which is key to improve human life. Similar to the adverse event report system, which can be used to evaluate drug–drug interactions, our proposed work can help identify the impact and relationship between lifestyle and diseases for better diagnosis. Therefore, big data mining methods are always essential for accurate scientific measurements.

Finally, the limitation related to the research results is that complete and accurate data are required for the analysis and a professional is needed to evaluate the rules created to determine relevance. In addition, it is mandatory to conduct benchmarking studies in different hospitals to identify and validate real cases to ensure the reliability of the system. Another obstacle to the practical application of data mining techniques in healthcare is the reliability of data, data sharing, and incorrect modeling leading to incorrect diagnosis. The complexity of data mining tools is another barrier to their use, as personnel must be trained to use them. For example, reliance on technology to interface with care can lead to confusion and frustration, especially among elderly and vulnerable patients. This can lead to confusion, poor understanding of treatment plans, or non-compliance with patient instructions.

## 7. Conclusions

Researchers internationally recognize the use of data mining to manage big data. Many publications support the use of data mining in the medical field. However, the aspect of the lifestyle history is missing when collecting data from patients for diagnosis in developing countries. Therefore, patient attributes, such as lifestyle patterns, including demographic factors, should also primarily addressed in applying data mining to clinical data. This will help in reducing diagnostic errors and increase administrative efficiency. A framework for HIMS with a knowledge base having some surplus characteristics of data analysis and reporting is proposed. Our study also presents important concerns and challenges in the health sector related to data analysis and reporting.

Various data analysis techniques were applied to patient (with eye disorder) data to retrieve useful hidden information from the database. The most popular pruning algorithm (Apriori) was applied to the dataset to find the most interesting and frequent patterns. This research work demonstrated that about 140 out of 1035 people have myopia at 22 years of age. It has also been found that no males are diagnosed with myopia, although the method applies to an equal proportion of males and females. It supports our view of lifestyle effects. The Rapid Miner data mining tool was used for data analysis purposes, and proof of finding frequent item-sets in EDCS was given using TANAGRA. Although the patterns found were interesting to study, the algorithm was implemented and served the intended

purpose. There are some limitations that will be addressed in the future. An effort can be made to dive deep in the dataset and find out the reason of having fatal diseases. Moreover, data mining can be modeled as lifestyle pattern mining, whereby rules can be created based on the provided lifestyle patterns.

**Author Contributions:** K.G.: Conceptualization, Data curation, Methodology, Writing—original draft, Software, Writing—review & editing, M.A.M.: Conceptualization, Data curation, Methodology, Writing—original draft, Supervision, Software, Writing—review & editing, Formal analysis, Project administration, Visualization, Investigation, S.M.M.: Project administration, Methodology, Visualization, Investigation, Writing—review & editing; Formal analysis, Funding acquisition, S.A.: Writing—review & editing; Formal analysis, Methodology, Investigation, Project administration, Funding acquisition, S.M.A.A.: Writing—review & editing; Formal analysis, Methodology, Investigation, Project administration, M.A.N.: Writing—review & editing; Formal analysis, Methodology, Investigation, Project administration. All authors have read and agreed to the published version of the manuscript.

**Funding:** This research received no external funding.

**Data Availability Statement:** Not applicable.

**Conflicts of Interest:** The authors declare no conflict of interest.

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
