# Peer review of "An Efficient Healthcare Data Mining Approach Using Apriori Algorithm: A Case Study of Eye Disorders in Young Adults"

_information, doi:10.3390/info14040203_

Round 1

Reviewer 1 Report

#Highlight the motivation, novelty and contribution clearly in the Introduction section. 

 #In related work section highlight the making and breaking of the previous study and summarize the same.

#Fig 1 is very generic and can be removed! Fig 2 needs to drawn using smart drawing software. All the fig.(s) are of very poor quality. 

#Write the experimental setup in detail.

#Discussion section needs major improvement. 

#Write limitations of the study in conclusion section.

I failed to find out any significant novelty in this study and presentation is also very poor.

Author Response

Dear Reviewer,

Thank you very much for your time and efforts to review our paper. I believe that the quality of our manuscript has been enhanced after considering your comments/ suggestion. please see the attached file for detailed responses to your comments.

Thanks,

BR,

Sheraz

Reviewer 2 Report

Title of Paper:    An Efficient Healthcare Data Mining Approach using Apriori Algorithm for Better Disease Diagnosis and Treatment.

Decision:  Conditionally Acceptable for publication after incorporating review comments as below.

Review comments:

1)     Refer to section “ References”: References are not sufficient. As such research in the last 3 years carried out in the related domain has not been discussed in detail in section 2, “literature review”. Please include/ discuss the latest work published in the related domain i.e. HIM Systems, algorithms for data mining, tools used for pattern discovery etc.  

2)     Refer to figures 2, 5,6, and 7, Text/ Font Size is not readable. The figures may be made readable.

3)     Caption for Figure.- 3 “ An example of publishing an existing document” is unsuitable.  This figure describes different phases of data analysis for eye disease patients. The caption may be made accordingly.

4)     Refer to section-2, Page 3/11, line 92: What are prons  & cons of apriori algorithm. Further, what is HaoYu Xie, concept, and how this is used in this research? Please explain in this paper.

5)     Refer to section 4: Please clarify/ explain Apriori algorithm presented at lines 208 to 229.

6)     Refer to Figure. 5 page 8/11: Captions of the figure may be “Scatter Chart of Eye diseases Vs Age”. Further, the Captions of Figure. 6 may be “Histogram for current age and diseases”.

7)     Refer to section 2: Apriori association algorithm has already been adopted in research papers at references 11, 13, and 14 for getting frequent items. How is your research different from the said previous research? Please highlight the novelty of this research.

8)     Name of the Journal for publication of the research may also be mentioned in the paper.

Author Response

(The authors gave the same response as above.)

Reviewer 3 Report

I would like to thank the authors for this work. Please consider the comments below in the next version.

(1)

The motivation behind the study should be clarified. For example, does the literature lack such studies?, or does the present study explore a new application of the Apriori algorithm?

(2)

In my view, the article would need to be re-positioned as a use case of rule mining in healthcare. This should have a reflection on the title, and discussion throughout the manuscript.

(3)

I am not clear what Section 3 (Proposed System Model…) presents or adds with respect to the use case under consideration. I am afraid I can’t see a particular aspect of novelty by this proposal in the context of Data Mining.

(4)

The related work should refer to studies that applied clustering as part of Data Mining applications in healthcare. For example:

https://doi.org/10.1145/3014812.3014874

https://doi.org/10.3390/app10249109

(5)

The quality of figures should be improved in general.

(6)

I find that the title is relatively vague. I recommend revising the title to give a more specific aspect about the use case under consideration.

(7)

Please discuss possible limitations related to the methodology.

(8)

Please confirm and mention in the manuscript whether the study acquired any form of ethical approval.

Author Response

(The authors gave the same response as above.)

Round 2

Reviewer 1 Report

Nice improvement. More revisions required:

#Fig 1, 2 needs improvement. Use smart drawing software (e.g. MS Visio) to do the same.

#Section 4 required additional references and more discussion. You may read and cite works like A survey of data mining and deep learning in bioinformatics. Journal of medical systems, 42, 1-20. (2018), Application of data mining techniques and data analysis methods to measure cancer morbidity and mortality data in a regional cancer registry: The case of the island of Crete, Greece. Computer methods and programs in biomedicine145, 73-83. (2017) etc.

#Improve resolution of fig 3and fig 6. 300 dpi will be good.

#How data mining can be deployed in other health care domain  (e.g. Large scale medical data mining for accurate diagnosis: A blueprint. Handbook of large-scale distributed Computing in smart healthcare, 157-176.) - needs to be highlighted in discussion section.

#It will be good to discuss the complexity of such techniques.

Author Response

(The authors gave the same response as above.)

Reviewer 3 Report

Thanks for accommodating the feedback, I have no further comments.

Author Response

(The authors gave the same response as above.)

Round 3

Reviewer 1 Report

Well revised.